# Effectiveness of Routine Measurement of Health-Related Quality of Life (HRQOL) in Improving Patient-reported Outcomes in Primary Care Patients with Chronic Knee and Back Problems – A Cluster Randomised Controlled Trial

Will H.G. Cheng[1‡], Kiki S. N. Liu[1‡], Amy P. P. Ng[1,2*], Carlos K. H. Wong[1,3,4], Calvin K. L. Or[5], Daniel Y. T. Fong[6], Jason P. Y. Cheung[7], David V. K. Chao[8,9], Welchie W. K. Ko[10], Eliza L. Y. Wong[11], Cindy L. K. Lam[1,2]

1 Department of Family Medicine and Primary Care, School of Clinical Medicine, Li Ka Shing Faculty of Medicine, the University of Hong Kong, Hong Kong SAR, China, 2 Department of Family Medicine, The University of Hong Kong-Shenzhen Hospital, Shenzhen, China, 3 Department of Infectious Disease Epidemiology, London School of Hygiene and Tropical Medicine, London, United Kingdom, 4 Laboratory of Data Discovery for Health (D24H), Hong Kong Science and Technology Park, Sha Tin, Hong Kong SAR, China, 5 Department of Industrial and Manufacturing Systems Engineering, Faculty of Engineering, the University of Hong Kong, Hong Kong SAR, China, 6 School of Nursing, Li Ka Shing Faculty of Medicine, The University of Hong Kong, Hong Kong SAR, China, 7 Department of Orthopaedics and Traumatology, School of Clinical Medicine, Li Ka Shing Faculty of Medicine, The University of Hong Kong, Hong Kong SAR, China, 8 Department of Family Medicine & Primary Health Care, United Christian Hospital, Kowloon East Cluster, Hospital Authority, Hong Kong SAR, China, 9 Department of Family Medicine & Primary Health Care, Tseung Kwan O Hospital, Kowloon East Cluster, Hospital Authority, Hong Kong SAR, China, 10 Family Medicine and Primary Healthcare Department, Queen Mary Hospital, Hong Kong West Cluster, Hospital Authority, Hong Kong SAR, China, 11 The Jockey Club School of Public Health and Primary Care, Faculty of Medicine, The Chinese University of Hong Kong SAR, Hong Kong SAR, China,

‡ Co-first authors
* amyppng@hku.hk

## Abstract

Chronic knee and back (knee/back) problems are common, painful, and disabling. Routine health-related quality of life (HRQOL) measurement may enhance doctor-patient communication and person-centred care by identifying unmet biopsychosocial needs, guiding personalized care, and encouraging engagement. We examined whether routine measurement and reporting of HRQOL using the electronic EuroQol 5-Dimension 5-Level (e-EQ-5D-5L) could improve HRQOL in participants allocated to the intervention (IG) or control (CG) groups. Participants were Chinese adult patients with a doctor-diagnosed knee/back problem with a scheduled follow-up visit within 12 months. The IG completed the e-EQ-5D-5L before each consultation at baseline and each scheduled follow-up visit, with reports available to their doctors during consultations. The primary outcome was the Western Ontario and McMaster Universities Osteoarthritis Index (WOMAC) score. Secondary outcomes were Patient Enablement Instrument-version 2 (PEI-2), Pain Rating Scale (PRS), and Short-form 6-Dimension

**Data availability statement:** All relevant data are within the manuscript and its Supporting Information files.

**Funding:** This study was funded by the Research Grant Council- General Research Fund, Hong Kong (RGC reference number 17100119 to CLKL). This publication received financial support from the EuroQol Group, which covered the open access publication fee. The funders had no role in study design, data collection and analysis, decision to publish, or preparation of the manuscript.

**Competing interests:** The authors have declared that no competing interests exist.

(SF-6D) health utility scores and management options. The effects of the intervention on the outcomes were assessed by generalised estimating equations (GEE) with linear function. 1200 participants were randomised to the IG (n = 595) and CG (n = 605). At 12 months, both groups reported higher (worse) WOMAC scores, with a greater increase in the IG than the CG (β = 2.43, p = 0.018). The IG reported higher (better) PEI-2 than the CG (β = 0.99, p = 0.010). There was no difference in PRS and SF-6D scores between groups. The IG received more oral medications, while the CG had more referrals to orthopaedic specialists. Routine measurement of HRQOL by the e-EQ-5D-5L did not improve HRQOL or pain, but was associated with better self-care enablement in primary care patients with chronic knee/back problems. HRQOL data may prompt primary care doctors to offer more conservative treatments before specialist referral, potentially easing the burden on secondary care.

## Author summary

Routine measurement of health-related quality of life (HRQOL) may facilitate patients' communication during consultations and facilitate doctors to be more patient-centred and enabling in the management. While the electronic-EuroQol 5-Dimension 5-Level (e-EQ-5D-5L) is a valid and feasible tool to measure HRQOL among patients with chronic knee/back problems in primary care in Hong Kong, the effectiveness of its routine measurement on improving the outcomes of these patients is unknown. This cluster randomised controlled trial found that routine measurement and reporting of HRQOL by the e-EQ-5D-5L was associated with better self-care enablement in primary care patients with chronic knee/back problems, although it did not alter the natural progression of the disease. The e-EQ-5D-5L data inform the doctors about the impact of the illness on the patients' quality of life, leading to more patient-centred management that addresses not only the pain but also the patient's stress and disability. The data may trigger the doctors to advise more self-care and consider a wider range of conservative treatments before specialist referrals. It has the potential to be applied as a routine measurement at the point-of-care to promote self-care enablement in patients with chronic musculoskeletal problems, as well as other chronic conditions, including mental health problems.

## Introduction

Musculoskeletal problems are a major global health burden [1,2]. In Hong Kong, they account for 9% of consultations in primary care, with chronic knee/back problems being the most common [3]. Chronic knee/back problems refer to a broad range of conditions, such as low back pain, osteoarthritis, tendinitis, or degenerative conditions. Chronic knee/back problems are disabling, which can limit patients' abilities to perform daily functions and impair their health-related quality of life (HRQOL) [4–6].

HRQOL assesses the impacts of health problems on the physical, psychological, and social aspects of a person's life [7]. Point-of-care routine measurement of HRQOL may facilitate doctor-patient communication during the consultation [8] and can lead to psychological benefits and reductions in mortality rates and emergency visits [9,10]. The availability of HRQOL data may trigger more patient-centred care by enabling shared decision-making [11–13]. Longitudinal data may inform doctors of the patients' ongoing needs and the effectiveness of management [14]. However, despite the potential benefits, routine measurement and reporting of HRQOL is rarely implemented in real clinical practice due to uncertainty on the clinical benefit and logistic barriers of time and workload [15], leaving its clinical impact unclear.

To facilitate routine HRQOL measurement in busy primary care clinics, our team developed an online platform for the administration and reporting of the electronic EuroQol 5-Dimension 5-Level (eEQ-5D-5L) HRQOL measure [16]. The EQ-5D-5L consists of 5 items and a visual rating scale (VAS), which produces a HRQOL profile, a utility index, and an overall health score [16,17]. It is available in many languages, including Chinese [17]. Previous studies have confirmed the validity, reliability, responsiveness, feasibility, and acceptability of the eEQ-5D-5L among Chinese patients with chronic knee/back problems in primary care [16,18].

We hypothesised that routine measurement and reporting of HRQOL with the eEQ-5D-5L could enhance patient self-care enablement, leading to better HRQOL and pain control, among patients with knee/back problems. Patient enablement, defined as "the extent to which a patient is capable of understanding and coping with his/her health issue" [19], is an important management strategy for patients with chronic diseases [20,21]. It is particularly important for progressive conditions such as chronic knee/back problems, where patient education and self-care are essential. By fostering greater understanding and coping ability, patient enablement could moderate the impact of pain on HRQOL [22] and serve as a link between HRQOL improvement and chronic disease self-management. We aimed to determine the effectiveness of routine measurement and reporting of the longitudinal e-EQ-5D-5L data in improving HRQOL, pain, and patient enablement among patients with chronic knee/back problems in primary care. We also explored how the intervention might affect doctors' management decisions.

## Method

### Ethics statement

This study was approved by the institutional review board of The University of Hong Kong/Hong Kong Hospital Authority Hong Kong West Cluster (UW18–270) and Hong Kong Hospital Authority Kowloon Central/Kowloon East Cluster (KC/KE-20–0070/ER-1). Formal written consent was obtained from all participants.

### Study design

This was a 12-month, single-blind, cluster randomised controlled trial (RCT). A table on the key information specified by the CONSORT guideline is shown in S1 Appendix [17]. Details on the study design, instruments, and procedures are available in the published study protocol [23]. No changes to methods were applied after trial commencement. The planned sample size was 1374 participants, including a 20% attrition rate. This study was approved by the institutional review board of The University of Hong Kong/Hong Kong Hospital Authority Hong Kong West Cluster (UW18–270) and Hong Kong Hospital Authority Kowloon Central/Kowloon East Cluster (KC/KE-20–0070/ER-1). This study was registered at the US ClinicalTrial.gov (NCT03609762) and the HKU clinical trials registry (HKUCTR-2418).

### Outcome measures

The primary outcome was the change in HRQOL, measured by the Western Ontario and McMaster Universities Osteoarthritis Index (WOMAC) total score. Secondary outcomes included changes in patient-reported patient enablement, pain level, overall health condition, and the Short-Form Six-Dimension (SF-6D) health utility scores. Doctors' management, including investigations, treatments, referrals, and patient self-care advice, at baseline and follow-up consultations, were exploratory outcomes.

## Randomisation

Randomisation was performed at the clinic level by a statistician (DYTF) who was not involved in recruitment and data collection. Six urban public primary care clinics, from two out of the seven geographic clusters organised by the Hospital Authority in Hong Kong, were randomised into the intervention group (IG) and control group (CG) by simple randomisation, with three clinics in each group. Service users in public primary care clinics in Hong Kong mainly included older adults, low-income individuals, and patients with chronic diseases [24]. For allocation concealment, we concealed the names of the clinics and allocated them with a number, ranging from 01 to 06, randomly, before randomisation to minimise bias. Follow-up outcome assessors who administered the questionnaire by telephone were blinded to the group allocation.

## Sample size calculation

The sample size calculation was based on the primary outcome of a difference in WOMAC total score between the intervention and control groups. A previous RCT study evaluating a rehabilitation programme integrating exercise, self-management and active coping strategies for patients with chronic knee pain showed that the average change in WOMAC total score at 6 months after baseline for the intervention and control groups was −3.4 (mean: 35; SD: 16 at 6 months) and −8 (mean: 30.4; SD: 17 at 6 months), respectively, indicating an effect size of 0.3 [25]. Using this, a minimum of 380 (190 in each group) participants were needed to achieve 80% power at 5% significant level, by two-sample t-test.

$$k = \frac{n_2}{n_1} = 1$$

$$n_1 = \frac{\left(\sigma_1^2 + \frac{\sigma_2^2}{K}\right)\left(z_{1-\frac{\alpha}{2}} + z_{1-\beta}\right)^2}{\Delta^2}$$

$$n_1 = \frac{\left(16^2 + \frac{16^2}{1}\right)(1.96 + 0.84)^2}{4.6^2}$$

$$n_1 = 190$$

$$n_2 = k * n_1 = 190$$

Note. $\Delta = |\mu2-\mu1| =$ absolute difference between two means, σ1, σ2 = variance of mean intervention and control group, $n_1$ = sample size for group 1, $n_2$ = sample size for group 2, α = probability of type I error, β = probability of type II error, z = critical Z value for a given α or β, and k = ratio of sample size for group #2 to group #1.

There were no studies that provided an intra-cluster correlation (ICC) coefficient for WOMAC scores, so we applied an ICC coefficient of 0.03 based on a review of pain outcome [25,26] to adjust for the clustering effect from six clinics, which increased the sample size to 1098 (549 patients per group). To allow for a 20% attrition rate, we planned to recruit 1374 (687 in each group) participants to the study.

$$N_{CRT} = (1 + (M-1)\,\rho) \times N_{IRT}$$

$$N_{CRT} = \left(1 + \left(\frac{380}{6} - 1\right)\ 0.03\right)\ \times\ 380$$

$$N_{CRT} = 1098.2 \approx 1098$$

Note. $\rho$ = ICC, $N_{CRT}$ = sample size needed for cluster randomised trial, $N_{IRT}$ = sample size needed for individual randomisation, and $M$ = cluster size.

## Patient recruitment

All Chinese adult patients with doctor-diagnosed chronic knee/back problems attending the study clinics were invited to participate. The inclusion criteria were: aged 18 years or above, had an existing symptomatic knee/back problem that was expected to last for one month or more, attending the clinic for a doctor consultation, and were scheduled for a follow-up visit in the same clinic within 12 months. The exclusion criteria were: had a life expectancy of less than 12 months, had current cancers, was too ill to complete a questionnaire, or was unable to communicate in Chinese. Eligible patients were either referred by their doctors or recruited by on-site trained research assistants.

## Intervention

Participants in the IG clinics completed the e-EQ-5D-5L before they consulted with the doctors at baseline and at each follow-up visit during the 12-month study period. A printed report summarising the longitudinal results on the EQ-5D-5L profile, utility, and VAS scores was generated for each participant to give to the doctor at each consultation. All doctors in the intervention clinics received a training session on the interpretation of the report on EQ-5D-5L HRQOL profile and VAS scores. Participants in the CG clinics received care as usual without completion of the e-EQ-5D-5L.

## Monitoring and reporting of adverse events

This study was considered to be a low-risk trial where both the intervention and control groups received their usual medical care. Rules for early stopping were not applicable as the clinicians involved in this study were not blinded to intervention allocation. Collection and assessment of reported adverse events and other unintended effects of the trial interventions or trial conduct were performed continuously, though none occurred throughout the trial. All unintended effects and adverse events were planned to be reported every six months to the Institutional Review Board (IRB) of the University of Hong Kong and Queen Mary Hospital.

## Data collection

After obtaining their written informed consents, all participants completed a structured-questionnaire on sociodemographic, treatment/healthcare services history (including current comorbidities, days of sick leave, and medication use), and patient-reported outcome measures (PROMs) by face-to-face interviews at baseline. Data on treatment/healthcare services history and PROMs were collected again by telephone interviews at 3, 6, and 12 months after baseline by other interviewers who were blinded to the group allocation. To ensure data consistency, all interviewers, including both face-to-face and telephone interviewers, had been trained by the same study investigators and were given a standardised script to follow verbally during interviews. The study flow diagram is shown in Fig 1.

The participating doctors in both groups completed a doctor-reported clinical case report form (CRF) after each consultation on the diagnosis, duration of the knee/back problem, management decision, a 5-point global rating scale (GRS) on the severity of the patients' condition, and a 7-point scale of global rating on the change in the patient's condition, ranging from -3 (much worse) to 3 (much better) [27].

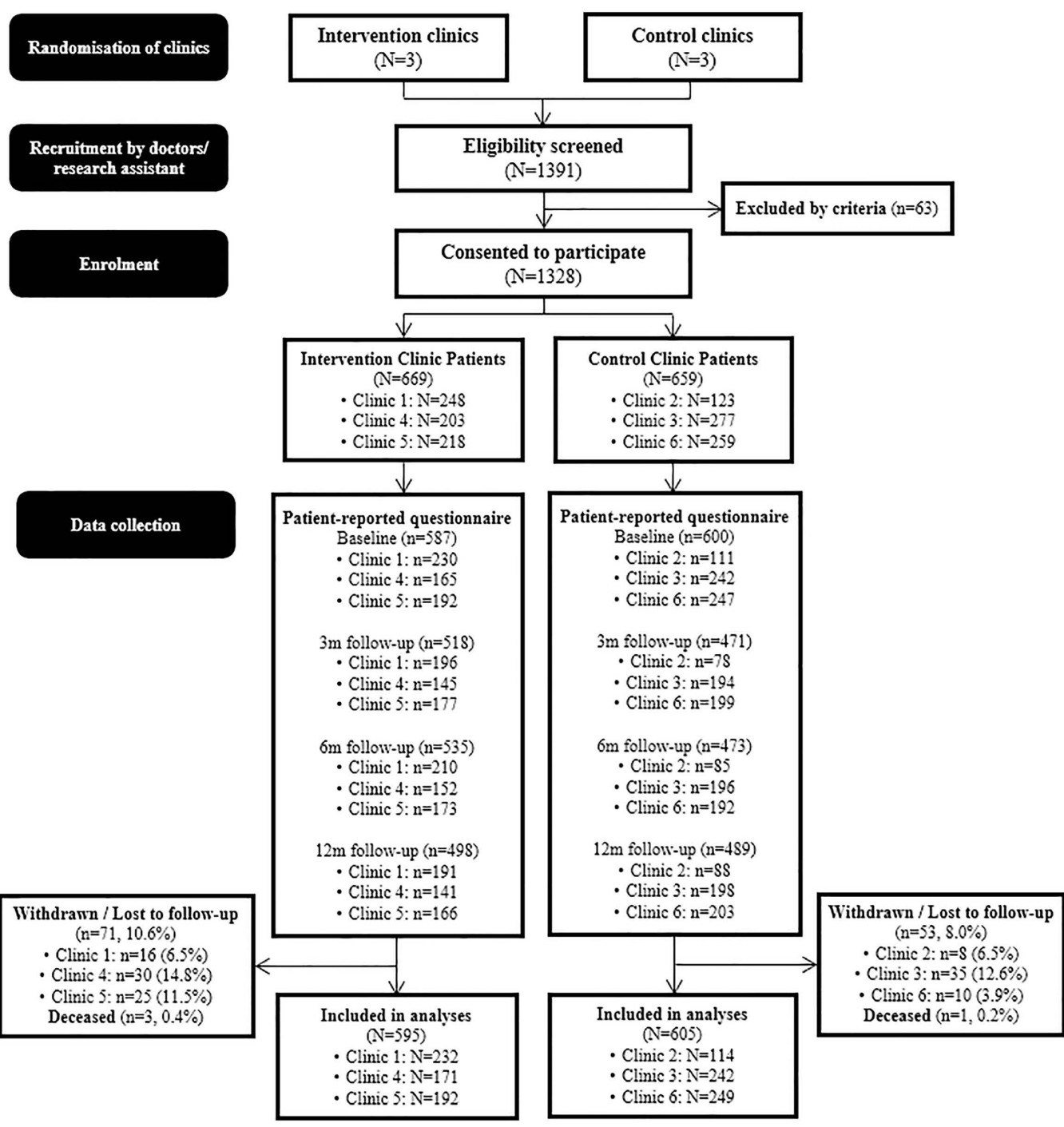

**Fig 1. Study flow diagram of the recruitment and data collection process.**

## Study instruments

The electronic Chinese (Hong Kong) EQ-5D-5L was only completed by participants in the IG. It comprises five items representing five HRQOL dimensions (mobility, self-care, usual activities, pain/discomfort, and anxiety/depression) and an e-tablet-adapted 100mm Visual Analogue Scale (EQ-VAS) on global health.

We administered the following PROMs to all participants in both groups at baseline and follow-ups:

i) The Chinese WOMAC is a 24-item condition-specific HRQOL measure on pain, stiffness, and difficulty in physical functioning [28–30]. Each item is rated on a 5-point Likert scale, ranging from 0 to 4, with higher scores indicating greater impairment. The total WOMAC score is the sum of the three domain scores [29] with a range from 0 to 96.

ii) The Chinese Patient Enablement Index-2 (PEI-2) is a 6-item measure of the patient's perceived ability in: i) coping with life, ii) understanding their illness, iii) coping with their illness, iv) keeping themselves healthy, v) helping themselves, and vi) feeling confident about their health. Each item is rated on a 5-point Likert scale, ranging from 1 to 5. The sum of the item scores gives the total PEI-2 score, ranging from 6 to 30, with higher scores indicating better enablement [31].

iii) The Pain Rating Scale (PRS) rates the severity of pain, ranging from 0 (no pain) to 10 (the worst pain) [32,33].

iv) The Chinese SF-6D is a 6-item preference-based measure of health on six dimensions (physical functioning, role limitations, social functioning, bodily pain, mental health, and vitality). The SF-6D responses are converted to a Hong Kong population-specific composite preference (utility) score from 0 (death) to 1 (perfect health) [34].

v) Global Rating Scale on change in health (GRS) is a 7-point scale measuring the patient's perception of any change in overall health condition, ranging from -3 (much worse) to 3 (much better).

## Statistical analysis

All data were analysed by intention to treat, in which participants who were randomised and had completed at least one repeat outcome measurement were included in the analysis. Analysis was restricted to participants with observed repeat outcome data (complete-case analysis for the study outcomes). Descriptive statistics described the socio-demographic, disease characteristics, PROMs, and the prevalence of self-reported treatment/healthcare services history at baseline. Participants with knee and/or back problems were analysed as a combined group. Paired-sample t-tests compared the within-group changes of the PROMs between baseline and follow-ups. We calculated the effect size of within-group change by dividing the difference by the group baseline standard deviation (SD), and the effect size of between-group changes by dividing the difference of differences by the overall baseline SD. A generalised estimating equation (GEE) with linear function and an exchangeable working correlation matrix was used to analyse the effectiveness of e-EQ-5D-5L on PROMs from baseline to follow-ups, adjusting for age, gender, the respective baseline PROM scores, global rating severity at baseline, diagnosis type, diagnosis duration, and number of comorbidities. Chi-square test was used to compare the difference in the prevalence of doctors' management decisions, and an independent sample t-test was used to assess the difference in doctor reported changes in GRS from baseline to follow-up consultations between groups. IBM SPSS Statistic V.27 and R 3.5.1 were used for the data analyses, taking a 5% level of significance.

## Results

1328 patients with chronic knee/back problems were recruited from June 2020 to February 2021. Of them, 1200 patients (595 in IG and 605 in CG) completed one or more follow-up assessments and were included in the analyses (Fig 1). The proportions of withdrawal or loss to follow-up were 10.6% in IG and 8.0% in CG, whilst the rate for each individual clinic ranged from 3.9% to 14.8%. The baseline characteristics between participants who had completed the study and those who had defaulted on follow-up are shown in the S1 Table. The defaulters were more likely to be younger and more

educated, have a back problem, higher household income, but no co-morbidity, and have received specialist care, surgery, and physiotherapy. Among those who had completed the study, over half (69.08%) were female, and the mean age of all participants was 68.80 (SD = 10.24) years old. 67.8% of them had a knee problem only, 18.8% had a back problem only, and 10.0% had both. Duration of the knee/back problems ranged from less than 1 year (17.0%) to more than 10 years (30.5%). Table 1 shows the baseline characteristics, including healthcare services utilisation in the previous 3 months, overall and by groups. There was no difference in PROM scores between the two groups at baseline. The IG reported a significantly higher prevalence of use of oral prescribed medication, while the CG reported a significantly higher prevalence of self-medication for their knee/back conditions. The CG reported higher prevalences, but statistically insignificant, of surgical treatment, physiotherapy, and specialist outpatient clinic appointments than the IG at baseline. The baseline characteristics of study participants by clinics are shown in S2 Table.

## Changes in PROMs

Table 2 shows the respective changes in PROMs from the baseline to 3-, 6-, and 12-months in both IG and CG. Table 3 presents the effects of the intervention (routine e-EQ-5D-5L measurement) on PROMs from baseline to 3, 6, and 12-month by GEE. The results of the sensitivity analysis by GEE with cluster robust standard errors to account for clustering effects are shown in S3 Table.

At the end of the study, the WOMAC total score increased in both groups. The increase in IG was significant (mean change: + 2.27, effect size 0.16, $p = 0.002$), but that in the CG was not (mean change: + 0.43, effect size 0.02, $p = 0.515$), indicating worsening in HRQOL over 12 months. The IG had a greater increase in the WOMAC total score (difference of difference = 1.84, effect size 0.13,.β = 2.43 [95%CI: 0.41, 4.44], p = 0.018) at 12-months, when compared to the CG (Table 3).

The PEI-2 total score decreased significantly in the CG (mean change: -1.28, effect size 0.35, $p < 0.001$), but the change was insignificant in the IG (mean change: -0.24, effect size 0.07, $p = 0.413$) after 12 months. The IG was associated with a higher mean PEI-2 score (difference of difference = 1.02, effect size 0.30, β = 0.99, [95%CI: 0.24, 1.75], p = 0.010) at 12 months when compared to the CG (Table 3).

Both groups had significantly lower PRS score at 3, 6, and 12 months from baseline (IG: mean change: -0.45 to -0.64, $p < 0.001$; CG: mean change: -0.31 to -0.77, $p < 0.001$), as shown in Table 2. The SF-6D health utility score in the IG decreased but had no significant change in the CG (IG: mean change: -0.03, $p < 0.001$; CG: mean change: 0.01, $p = 0.085$). There was no significant association in change of SF-6D and PRS scores with the IG or CG groups (SF-6D: β = -0.010 [95%CI: -0.03, 0.01], p = 0.205; PRS: β = 0.070 [95%CI: -0.29, 0.43], p = 0.722) after the study (Table 3).

The sensitivity analysis by GEE with correction of clustering effect showed similar trends, although the positive effect of intervention on PEI-2 score did not reach statistical significance.

## Doctors' management and global rating

Table 4 shows the prevalence of the doctors' management options and the doctor-reported global rating on disease severity in the baseline and follow-up consultations in both groups. The IG were more likely to receive advice on home exercise at the baseline and the 1st follow-up consultations, and advice on braces in follow-ups. They were also more likely to be prescribed oral medications in follow-ups. The CG was more likely to be referred to physiotherapy, occupational therapy, and prosthetic and orthotic specialists in the baseline consultation. The patterns of treatments reported by the patients were similar to those recorded by the doctors, supporting the reliability of the results. The prevalence and odds ratios of patient-reported treatments by groups are shown in S4 Table.

The mean of doctor-reported GRS scores in the IG was significantly higher than those in CG in the 2nd and 3rd follow-up consultations ($p = 0.006$, $p = 0.002$). There was a significantly higher proportion of patients rated to have improved in GRS reported by the doctors in the CG ($p = 0.033$) than in the IG in the 2nd follow-up consultation.

**Table 1. Baseline characteristics of study participants (N = 1200).**

| | All participants (N = 1200) | Intervention (N = 595) | Control (N = 605) | P-value |
|---|---|---|---|---|
| | | n (%)/ mean (SD) | | |
| **Socio-Demographic** | | | | |
| Sex | | | | 0.536 |
| Male | 371 (30.9) | 179 (30.1) | 192 (31.7) | |
| Female | 829 (69.1) | 416 (69.9) | 413 (68.3) | |
| Age (years old) | 68.80 ± 10.24 | 68.39 (±10.30) | 69.19 (±10.18) | 0.179 |
| Education | | | | <0.001* |
| Primary or less | 635 (52.9) | 347 (58.3) | 288 (47.6) | |
| Secondary | 463 (38.6) | 208 (35.0) | 255 (42.1) | |
| Tertiary or above | 100 (8.3) | 40 (6.7) | 60 (9.9) | |
| Marital status | | | | 0.081 |
| Never married | 77 (6.4) | 41 (6.9) | 36 (6.0) | |
| Married | 923 (76.9) | 440 (73.9) | 483 (79.8) | |
| Separated/ divorced | 43 (3.6) | 24 (4.0) | 19 (3.1) | |
| Widowed | 151 (12.6) | 88 (14.8) | 63 (10.4) | |
| Occupation | | | | <0.001* |
| Unemployed/Retired | 627 (52.3) | 274 (46.1) | 373 (61.7) | |
| Homemaker | 282 (23.5) | 191 (32.1) | 91 (15.0) | |
| Labour worker | 126 (10.5) | 50 (8.4) | 76 (12.6) | |
| Clerical Worker | 48 (4.0) | 21 (3.5) | 27 (4.5) | |
| Professional/Manager | 27 (2.3) | 16 (2.7) | 11 (1.8) | |
| Others | 67 (5.6) | 43 (7.2) | 24 (4.0) | |
| Household monthly income (HKD)[†] | | | | 0.107 |
| 0-$9999 | 604 (50.3) | 274 (46.1) | 330 (54.5) | |
| $10000-19999 | 156 (13.0) | 71 (11.9) | 85 (14.0) | |
| $20000-29999 | 76 (6.3) | 43 (7.2) | 33 (5.5) | |
| >$29999 | 112 (9.3) | 51 (8.6) | 61 (10.1) | |
| Smoking | | | | 0.876 |
| Non-smoker | 1068 (89.0) | 533 (89.6) | 535 (88.4) | |
| Ex-smoker | 85 (7.1) | 41 (6.9) | 44 (7.3) | |
| Current Smoker | 43 (3.6) | 20 (3.4) | 23 (3.8) | |
| Alcohol drinking | | | | 0.293 |
| Non-drinker | 1034 (86.2) | 508 (85.4) | 526 (86.9) | |
| Ex-drinker | 46 (3.8) | 28 (4.7) | 18 (3.0) | |
| Current drinker[§] | 117 (9.8) | 59 (9.9) | 58 (9.6) | |
| **Disease characteristics** | | | | |
| Diagnosis of musculoskeletal problem | | | | 0.200 |
| Back only | 266 (22.2) | 135 (22.7) | 131 (21.7) | |
| Knee only | 814 (67.8) | 392 (65.9) | 422 (69.8) | |
| Both | 120 (10.0) | 68 (11.4) | 52 (8.6) | |
| Duration | | | | 0.242 |
| <1 year | 201 (16.8) | 94 (15.8) | 107 (17.7) | |
| 1-5 years | 398 (33.2) | 210 (35.3) | 188 (31.1) | |
| 5-10 years | 223 (18.6) | 117 (19.7) | 106 (17.5) | |
| >10 years | 360 (30.0) | 168 (28.2) | 192 (31.7) | |

*(Continued)*

| | All participants (N = 1200) | Intervention (N = 595) | Control (N = 605) | P-value |
|---|---|---|---|---|
| | | n (%)/ mean (SD) | | |
| Total number of comorbidities | | 1.39 (±0.95) | 1.35 (±0.82) | 0.462 |
| Comorbidities | | | | |
| No chronic disease | 107 (8.9) | 67 (11.3) | 30 (5.0) | <0.001* |
| Heart disease | 100 (8.3) | 55 (9.2) | 45 (7.4) | 0.258 |
| Hypertension | 900 (75.0) | 427 (71.8) | 473 (78.2) | 0.010* |
| Stroke | 37 (3.1) | 21 (3.5) | 16 (2.6) | 0.375 |
| Diabetes | 324 (27.0) | 180 (30.3) | 144 (23.8) | 0.012* |
| Lung disease | 27 (2.3) | 18 (3.0) | 9 (1.5) | 0.073 |
| Mental illness | 50 (4.2) | 24 (4.0) | 26 (4.3) | 0.819 |
| Kidney disease | 17 (1.4) | 12 (2.0) | 5 (0.8) | 0.081 |
| Other joint problem | 162 (13.5) | 72 (12.1) | 90 (14.9) | 0.160 |
| Cancer | 24 (2.0) | 16 (2.7) | 8 (1.3) | 0.091 |
| Other diseases | 188 (15.7) | 61 (10.3) | 127 (21.0) | <0.001* |
| Doctor-reported GRS score (range 0–4) | 2.44 ± 0.64 | 2.48 (±0.65) | 2.40 (±0.625) | 0.051 |
| WOMAC total score (range 0–96) | 20.62 ± 14.42 | 20.08 (±13.84) | 21.14 (±14.97) | 0.200 |
| WOMAC pain score (range 0–20) | 5.12 ± 3.44 | 4.98 (±3.32) | 5.25 (±3.55) | 0.153 |
| WOMAC stiffness score (range 0–8) | 1.62 ± 1.67 | 1.58 (±1.61) | 1.66 (±1.72) | 0.162 |
| WOMAC function score (range 0–68) | 13.88 ± 10.77 | 13.52 (±10.22) | 14.23 (±11.27) | 0.094 |
| PEI-2 total score (range 6–30) | 21.61 ± 3.44 | 21.66 (±3.15) | 21.56 (±3.70) | 0.622 |
| SF-6D utility score (range 0.291 to 1) | 0.72 ± 0.15 | 0.72 (±0.15) | 0.71 (±0.14) | 0.070 |
| PRS score (range 0–10) | 5.40 ± 2.30 | 5.49 (±2.43) | 5.31 (±2.17) | 0.169 |
| **Prevalence of self-reported treatment/healthcare service utilized in the 3 months prior to start of study** | | | | |
| Oral prescribed medication | 657 (54.8) | 348 (58.5) | 309 (51.1) | 0.009* |
| Physiotherapy | 121 (10.1) | 52 (8.7) | 69 (11.4) | 0.121 |
| Occupational therapy | 11 (0.9) | 4 (0.7) | 7 (1.2) | 0.377 |
| Surgery | 26 (2.2) | 9 (1.5) | 17 (2.8) | 0.123 |
| Sick leaves | 28 (2.3) | 14 (2.4) | 14 (2.3) | 0.964 |
| Self-medication | 308 (25.7) | 113 (19.0) | 195 (32.2) | <0.001* |
| Accident & Emergency visits | 15 (1.3) | 7 (1.2) | 8 (1.3) | 0.820 |
| Specialist outpatient visits | 83 (6.9) | 38 (6.4) | 45 (7.4) | 0.473 |
| Hospital admission | 17 (1.4) | 8 (1.3) | 9 (1.5) | 0.834 |

## Discussion

### Summary

This study found that HRQOL measured by WOMAC worsened over 12 months in both intervention and control groups, and the change was significantly greater in the IG than the CG. However, the effect sizes of within-group changes (IG:0.16; CG: 0.02) and that of between-group difference in changes (0.13) of the WOMAC total score were smaller than the range of 0.2 to 0.5 associated with a minimal clinically important difference (MCID) for PROM reported in the literature [35,36]. Therefore, the difference might not be clinically important.

Although routine measurement and reporting of HRQOL by the e-EQ-5D-5L did not lead to better improvements in HRQOL or pain in our study, the observed difference in patient enablement (PEI-2 score) suggested a possible

**Table 2. Comparisons of changes from baseline to 3, 6 and 12-month in patient-reported outcomes by intervention and control groups (N=1200).**

| | Intervention | | | | | | Control | | | | | |
|---|---|---|---|---|---|---|---|---|---|---|---|---|
| | Baseline and 3 months follow-up (n=511) | Paired Difference (p-value) | Baseline and 6 months follow-up (n=528) | Paired Difference (p-value) | Baseline and 12 months follow-up (n=492) | Paired Difference (p-value) | Baseline and 3 months follow-up (n=468) | Paired Difference (p-value) | Baseline and 6 months follow-up (n=471) | Paired Difference (p-value) | Baseline and 12 months follow-up (n=486) | Paired Difference (p-value) |
| WOMAC total score | 19.92±13.53; 24.37±17.11 | 4.45±13.33 (<0.001*) | 19.85±13.62; 22.81±17.11 | 2.96±15.21 (<0.001*) | 19.87±13.54; 22.15±17.81 | 2.27±16.54 (0.002*) | 20.41±14.67; 23.46±16.35 | 3.05±14.94 (<0.001*) | 20.55±15.04; 22.01±16.75 | 1.46±14.98 (0.035*) | 20.78±14.96; 20.35±17.20 | 0.43±14.60 (0.515) |
| WOMAC pain score | 4.95±3.25; 5.61±3.86 | 0.67±3.31 (<0.001*) | 4.97±3.28; 5.33±3.88 | 0.35±3.85 (0.036*) | 4.90±3.17; 5.55±4.24 | 0.65±4.10 (<0.001*) | 5.09±3.45; 5.33±3.72 | 0.24±3.47 (0.140) | 5.21±3.59; 4.92±3.82 | -0.30±3.79 (0.089) | 5.20±3.55; 5.07±3.99 | -0.12±3.90 (0.485) |
| WOMAC stiffness score | 1.59±1.61; 1.98±1.73 | 0.39±1.85 (<0.001*) | 1.56±1.61; 1.78±1.69 | 0.22±1.85 (0.004*) | 1.58±1.61; 1.57±1.73 | -0.01±1.91 (0.944) | 1.60±1.73; 1.81±1.76 | 0.22±1.90 (0.014*) | 1.58±1.70; 1.73±1.70 | 0.16±1.95 (0.081) | 1.63±1.71; 1.63±1.77 | -0.00±1.95 (1.000) |
| WOMAC functional score | 13.38±9.98; 16.79±12.62 | 3.41±10.14 (<0.001*) | 13.31±10.08; 15.72±12.63 | 2.41±11.52 (<0.001*) | 13.40±10.04; 15.02±13.03 | 1.62±12.37 (0.004*) | 13.73±11.00; 16.32±12.11 | 2.59±10.69 (<0.001*) | 13.77±11.28; 15.36±12.56 | 1.59±11.28 (0.002*) | 13.96±11.19; 13.64±12.79 | -0.32±11.02 (0.524) |
| PEI-2 total score | 21.60±3.11; 19.44±3.82 | -2.16±4.25 (<0.001*) | 21.69±3.13; 19.96±3.90 | -1.73±4.22 (<0.001*) | 21.68±3.15; 21.44±5.76 | -0.24±5.98 (0.413) | 21.84±3.55; 19.65±3.99 | -2.19±4.55 (<0.001*) | 21.94±3.58; 20.04±3.85 | -1.90±1.28 (<0.001*) | 21.85±3.63; 20.57±4.30 | -1.28±4.90 (<0.001*) |
| SF-6D utility score | 0.72±0.15; 0.68±0.15 | -0.04±0.14 (<0.001*) | 0.73±0.15; 0.69±0.15 | -0.04±0.15 (<0.001*) | 0.73±0.15; 0.70±0.16 | -0.03±0.16 (<0.001*) | 0.71±0.14; 0.70±0.15 | 0.02±0.13 (0.008*) | 0.71±0.14; 0.71±0.15 | 0.01±0.14 (0.330) | 0.71±0.14; 0.70±0.15 | 0.01±0.14 (0.085) |
| PRS score | 5.51±2.42; 5.05±2.32 | -0.45±2.49 (<0.001*) | 5.53±2.48; 4.94±2.37 | -0.59±2.71 (<0.001*) | 5.46±2.36; 4.82±2.60 | -0.64±3.01 (<0.001*) | 5.16±2.15; 4.86±2.37 | -0.31±2.24 (0.003*) | 5.21±2.16; 4.68±2.43 | -0.53±2.41 (<0.001*) | 5.25±2.16; 4.49±2.43 | -0.77±2.53 (<0.001*) |

WOMAC=The Western Ontario and McMaster Universities Osteoarthritis Index (higher score indicates more limitation); SD=standard deviation; SF-6D=Short-Form Six-Dimension (higher score indicates better health); PEI-2=Patient Enablement Instrument-version 2 (higher score indicates more enabled); PRS=Pain Rating Scale (higher score indicates more pain).

* Significant at p<0.05 by paired-sample t-tests.

**Table 3. Effects of intervention by routine e-EQ5D5L measurement on changes in patient-reported outcomes from baseline to 3, 6 and 12-month (N=1200).**

| Patient-reported outcomes | WOMAC total score (n=1198) | | | | SF6D utility score (n=1194) | | | | PEI-2 total score (n=1189) | | | | PRS (n=1199) | | | |
|---|---|---|---|---|---|---|---|---|---|---|---|---|---|---|---|---|
| | Unadjusted | | Adjusted | | Unadjusted | | Adjusted | | Unadjusted | | Adjusted | | Unadjusted | | Adjusted | |
| | β (95 CI) | p-value | β§ (95 CI) | p-value | β (95 CI) | p-value | β§ (95 CI) | p-value | β (95 CI) | p-value | β§ (95 CI) | p-value | β (95 CI) | p-value | β§ (95 CI) | p-value |
| **Time (vs Baseline) * Intervention (vs Control)** | | | | | | | | | | | | | | | | |
| 12 months | 2.32 (0.32, 4.33) | 0.023* | 2.43 (0.41, 4.44) | 0.018* | -0.01 (-0.03, 0.01) | 0.237 | -0.01 (-0.03, 0.01) | 0.205 | 0.92 (0.17, 1.67) | 0.017* | 0.99 (0.24, 1.75) | 0.010* | 0.05 (-0.31, 0.41) | 0.771 | 0.07 (-0.29, 0.43) | 0.722 |
| 6 months | 1.30 (-0.63, 3.23) | 0.187 | 1.31 (-0.63, 3.25) | 0.185 | -0.03 (-0.05, -0.01) | 0.002* | -0.03 (-0.05, -0.01) | 0.002* | -0.13 (-0.70, 0.44) | 0.651 | -0.04 (-0.60, 0.53) | 0.902 | -0.12 (-0.45, 0.20) | 0.461 | -0.13 (-0.46, 0.20) | 0.426 |
| 3 months | 1.64 (-0.13, 3.41) | 0.069 | 1.48 (-0.29, 3.24) | 0.100 | -0.03 (-0.05, -0.01) | 0.003* | -0.03 (-0.05, -0.01) | 0.005* | -0.19 (-0.79, 0.40) | 0.525 | -0.13 (-0.73, 0.46) | 0.666 | -0.07 (-0.38, 0.23) | 0.642 | -0.10 (-0.40, 0.21) | 0.530 |

e-EQ5D5L = electronic-EuroQol 5-Dimension 5-Level; PEI-2 = Patient Enablement Instrument-version 2 (higher score indicates more enabled); PRS = Pain Rating Scale (higher score indicates more pain); SD = standard deviation; SF-6D = Short-Form Six-Dimension (higher score indicates higher health utility); WOMAC = The Western Ontario and McMaster Universities Osteoarthritis Index (higher score indicates more limitation); 95 CI = 95% of the confidence interval.

Notes.§ Analysis by a generalised estimating equation (GEE) with linear function and an exchangeable working correlation matrix, adjusted by age, gender, the respective baseline scores, global rating severity of baseline, diagnosed type, diagnosed duration, and number of comorbidities.

* Significant at p < 0.05.

**Table 4. Doctors' management and global rating on disease condition at baseline and follow-up consultations by groups (N = 1200).**

| | Baseline consultation | | | 1st follow-up consultation (approx. 3–5 months after the baseline consultation) | | | 2nd follow-up consultation (approx. 6–8 months after the baseline consultation) | | | 3rd follow-up consultation (approx. 9–11 months after the baseline consultation) | | |
|---|---|---|---|---|---|---|---|---|---|---|---|---|
| | Intervention (N = 595) | Control (N = 605) | p-value | Intervention (N = 595) | Control (N = 605) | p-value | Intervention (n = 464) | Control (n = 499) | p-value | Intervention (n = 292) | Control (n = 301) | p-value |
| | n (%)/ mean (SD) | | | n (%)/ mean (SD) | | | n (%)/ mean (SD) | | | n (%)/ mean (SD) | | |
| **Doctor-prescribed treatment** | | | | | | | | | | | | |
| **Investigation ordered** | | | | | | | | | | | | |
| X-ray | 42 (7.1) | 53 (8.8) | 0.275 | 9 (1.5) | 6 (1.0) | 0.434 | 5 (1.1) | 9 (1.8) | 0.337 | 2 (0.7) | 4 (1.3) | 0.429 |
| **Treatment prescribed** | | | | | | | | | | | | |
| Topical medication | 215 (36.1) | 218 (36.0) | 0.971 | 188 (31.6) | 190 (31.4) | 0.943 | 135 (22.7) | 147 (24.3) | 0.511 | 79 (13.3) | 84 (13.9) | 0.759 |
| Oral medication | 298 (50.1) | 258 (45.6) | 0.010* | 263 (44.2) | 184 (30.4) | <0.001* | 196 (32.9) | 150 (24.8) | 0.002* | 120 (20.2) | 84 (13.9) | 0.004* |
| Paracetamol | 241 (40.5) | 192 (31.7) | 0.002* | 210 (35.3) | 147 (24.7) | <0.001* | 159 (34.3) | 115 (23.3) | <0.001* | 93 (32.0) | 70 (23.5) | 0.022* |
| NSAID | 73 (12.3) | 72 (11.9) | 0.845 | 64 (10.8) | 44 (7.4) | 0.043* | 40 (8.6) | 35 (7.1) | 0.376 | 29 (10.0) | 14 (4.7) | 0.014* |
| Tramadol | 37 (6.2) | 20 (3.3) | 0.018* | 34 (5.7) | 17 (2.9) | 0.015* | 21 (4.5) | 14 (2.8) | 0.163 | 13 (4.5) | 8 (2.7) | 0.243 |
| **Referral** | 43 (7.2) | 74 (12.2) | 0.003* | 17 (2.9) | 35 (5.8) | 0.013* | 24 (4.0) | 18 (3.0) | 0.319 | 7 (1.2) | 9 (1.5) | 0.638 |
| Physiotherapy | 29 (4.9) | 50 (8.3) | 0.018* | 15 (2.5) | 20 (3.4) | 0.394 | 17 (3.7) | 11 (2.2) | 0.187 | 4 (1.4) | 4 (1.3) | 0.973 |
| Occupational therapy | 2 (0.3) | 9 (1.5) | 0.036* | 0 (0.0) | 3 (0.5) | 0.083 | 1 (0.8) | 4 (0.8) | 0.202 | 0 (0.0) | 2 (0.7) | 0.162 |
| Prosthetic and Orthotic | 0 (0.0) | 10 (1.7) | 0.002* | 0 (0.0) | 2 (0.3) | 0.157 | 0 (0.0) | 1 (0.2) | 0.332 | 1 (0.3) | 0 (0.0) | 0.311 |
| Orthopaedics | 10 (1.7) | 14 (2.3) | 0.451 | 3 (0.5) | 10 (1.7) | 0.051 | 7 (1.5) | 4 (0.8) | 0.310 | 7 (1.2) | 9 (1.5) | 0.638 |
| **Patient self-care advice** | | | | | | | | | | | | |
| None | 158 (27.9) | 195 (35.1) | 0.009* | 241 (49.6) | 286 (60.0) | 0.001* | 216 (56.5) | 247 (61.1) | 0.191 | 162 (67.5) | 170 (66.4) | 0.796 |
| Ice pack | 43 (7.7) | 14 (2.6) | <0.001* | 32 (6.6) | 11 (2.3) | 0.001* | 18 (4.8) | 6 (1.5) | 0.009* | 11 (4.7) | 1 (0.4) | 0.002* |
| Heat pack | 70 (12.5) | 152 (28.6) | <0.001* | 31 (6.4) | 81 (17.1) | <0.001* | 23 (6.1) | 31 (7.8) | 0.341 | 8 (3.4) | 21 (8.3) | 0.024* |
| Brace | 26 (4.6) | 8 (1.5) | 0.003* | 22 (4.5) | 5 (1.1) | 0.001* | 16 (4.2) | 3 (0.8) | 0.002* | 14 (6.0) | 0 (0.0) | <0.001* |
| Home Exercise | 321 (57.2) | 197 (37.1) | <0.001* | 191 (39.5) | 117 (24.7) | 0.001* | 128 (33.9) | 121 (30.6) | 0.325 | 55 (23.6) | 64 (25.3) | 0.665 |
| **Global rating severity score** | 2.48 ± 0.65 | 2.40 ± 0.63 | 0.051 | 2.31 ± 0.68 | 2.34 ± 0.65 | 0.413 | 2.34 ± 0.68 | 2.22 ± 0.63 | 0.006* | 2.29 ± 0.69 | 2.11 ± 0.64 | 0.002* |
| **Changes in GRS compared to the previous consultation** | | | | | | 0.071 | | | 0.033* | | | 0.240 |
| Improved | N/A | N/A | N/A | 146 (28.1) | 140 (26.1) | | 75 (20.1) | 112 (26.8) | | 54 (22.8) | 81 (29.2) | |
| Unchanged | N/A | N/A | N/A | 304 (58.5) | 296 (55.1) | | 212 (56.8) | 233 (55.7) | | 132 (55.7) | 142 (51.3) | |
| Worsened | N/A | N/A | N/A | 70 (13.5) | 100 (18.6) | | 86 (23.1) | 73 (17.5) | | 51 (21.5) | 53 (19.1) | |
| **Changes in GRS score** | N/A | N/A | N/A | 0.20 ± 0.91 | 0.26 ± 0.92 | 0.327 | 0.21 ± 1.02 | 0.30 ± 0.86 | 0.143 | 0.17 ± 1.01 | 0.31 ± 0.76 | 0.062 |

Approx. = approximately; GRS = Global Rating Scale on change in health condition; N/A = Not applicable; NSAID = Non-Steroidal Anti-Inflammatory Drug; SD = Standard Deviation.

Notes.* Significant at p < 0.05 by chi square test.

enhancement in patients' capacity to understand and manage their condition. The PEI-2 is a validated, reliable, and responsive measure, capable of detecting changes over time. [31] The effect size of the between-group difference in changes in PEI-2 score after 12 months was moderate (0.30) [37], which falls within the MCID range of 0.2 to 0.5 for PROM reported in the literature [35,36]. A point to note is that the sensitivity analysis of GEE with correction of clustering effect showed the correlation between the intervention and change in PEI-2 score did not reach statistical significance; further studies are needed to confirm the benefit.

The findings suggest that routine measurement and reporting of HRQOL can promote self-care enablement among patients with chronic knee/back problems in primary care. The e-EQ-5D-5L data can inform doctors about the psychosocial problems of the patients, leading to more patient-centred management that addresses not only the pain but also the patient's stress and disability. Moreover, the information might encourage more conservative treatments, such as exercise and lifestyle modifications, before referrals for invasive procedures.

## Change in HRQOL in patients with chronic knee/back problems

Several contextual and clinical factors may explain why our intervention did not improve HRQOL or pain in the participants. First, chronic knee and back conditions are predominantly degenerative, characterized by irreversible structural changes and progressive functional decline. These pathophysiological constraints limit the potential for PROM feedback alone to produce measurable improvements in physical outcomes. Second, a higher proportion of CG patients received referrals to orthopaedic specialists and physiotherapy early in the study, which likely resulted in more intensive physical interventions, including surgery, that could have influenced HRQOL trajectories. Furthermore, the study was conducted during the quarantine-imposed period of the COVID-19 pandemic when outdoor activities were restricted. A reduction in mobility is well-documented to exacerbate musculoskeletal symptoms and impair HRQOL [38,39], and was observed across both intervention and control groups. Previous studies have shown that routine PROM feedback often improves mental health outcomes, such as reducing anxiety or depressive symptoms, but the effect on physical outcomes like pain or HRQOL is less consistent. A review on the effectiveness of using HRQOL in practice routinely to improve outcomes of patient care concluded that such measurement mainly facilitated the detection and early management of psychological problems rather than direct changes in physical functioning or disease progression. Our trial targeted chronic musculoskeletal conditions, where irreversible structural changes and the progressive nature of pain and functional decline likely limit the capacity of PROM feedback to produce measurable improvements in HRQOL or pain.

## Patient enablement and patient activation

The benefit on patient enablement, despite the lack of significant improvement in HRQOL and pain, can be explained through behavioural and communication pathways. Hickman et al (2022) explained that patient enablement involves the initial acquisition of skills and knowledge to engage in healthcare, and only once patients have acquired these skills can they have patient empowerment [40]. The eEQ-5D-5L report on HRQOL can trigger doctor-patient communication and shared decision making on strategies to cope with the chronic knee/back problem better. Research shows that shared decision making helps incorporate patients' individual context, values, goals, and preferences into treatment planning. Effective communication and collaboration between clinicians and patients can lead to better outcomes, greater satisfaction, and reduced healthcare costs [40]. Additionally, it may make psychosocial needs more visible, prompting discussions that go beyond symptom control to include management of stress and disability. The reflection on HRQOL may also enhance patients' understanding of their health status, fostering greater confidence and motivation for self-care—concepts aligned with the theory of patient activation [41].Higher levels of activation are consistently associated with better self-management, improved clinical outcomes, and reduced healthcare utilization [42,43]. Patient activation and shared decision-making are central to contemporary models of patient-centred care and are precursors of behaviour change [40]. Together, these mechanisms—enablement, activation, and shared decision-making—provide a conceptual basis for

understanding how the routine reporting of HRQOL can enhance patient engagement and self-care, which are particularly relevant for patients with chronic musculoskeletal conditions.

### Strengths and limitations

This study employed a cluster RCT design to mitigate contamination bias, involving a large sample across multiple primary care clinics, enhancing the potential generalizability to other public primary care settings in Hong Kong. A few limitations of this study exist. First, only six clinics were included, and we did not perform any stratification or matching to balance clinic characteristics between groups, which might have led to bias despite randomization. Second, we only recruited Chinese patients from public primary care clinics. Such results may not be generalizable to other ethnic groups or patients managed in private primary care. Third, self-reported PROMs are susceptible to reporting biases, including recall bias. Fourth, the study was conducted during COVID-19 restrictions, which may have influenced patients' physical activity, access to healthcare, and health behaviours, potentially confounding outcomes. Additionally, the change from face-to-face at baseline to telephone interviews at follow-up could have introduced inconsistency in response, but we believe the bias should be small since the interviewers were trained and previous studies have confirmed inter-rater reliability between face-to-face and telephone interview modes [44,45].

### Implications for research and practice

Our previous study found that our e-EQ-5D-5L tool was feasible and acceptable in real clinical practice in Hong Kong 2026/3/31, but it was met with several challenges, including doctors feeling it took extra time in a short consultation and elderly patients having difficulty using and understanding the electronic tool. A recent systematic review on electronic patient-reported outcome measures (ePROMs) in chronic disease management within primary care found mixed effectiveness [46–48]. Key barriers included digital literacy gaps and workflow integration challenges, whereas facilitators involved strong patient-provider relationships and personalized feedback. These findings suggest that contextual factors, rather than technology alone, would need to be overcome before implementation into real practice. Further mixed-methods research that includes qualitative evaluations is needed to explore how, why, and why not PROM feedback influences patient and clinician behaviors, perceptions, and decision-making. Moreover, conducting cost-effectiveness analyses is crucial to determine whether routine HRQOL measurement provides value relative to its costs, including staff time, helping policymakers and healthcare providers make informed decisions

The integration of PROM feedback into clinical workflows can be enhanced through embedding PROM data within electronic health records, enabling real-time access and longitudinal tracking during consultations, as this would reduce the manpower, time, and resources spent printing the reports, which are important sustainability considerations. Developing clinician decision-support tools that utilize PROM data may facilitate personalized management plans, promote shared decision-making, and improve patient engagement. Additionally, incorporating PROMs into patient portals or mobile health applications can empower patients to self-monitor and actively participate in their care outside of clinical encounters. Systematic evaluation of these integration approaches will help identify effective models for implementation in primary care.

### Conclusion

While our study showed that routine measurement and reporting of HRQOL using the e-EQ-5D-5L did not lead to significant improvements in HRQOL or pain among patients with chronic knee and/or back problems, the observed benefit in patient enablement suggests that PROM feedback may play a role in enhancing patients' capacity for self-care and engagement. Integrating PROMs into primary care could support more personalized, patient-centred management strategies. However, given the modest effects observed and the study's limitations, further studies are required to confirm the benefit and establish the implementation model. Eventually, policymakers and primary care providers may consider

PROMs as part of broader strategies to promote patient empowerment, shared decision-making, and tailored care pathways, ultimately aiming to improve long-term health outcomes in a resource-efficient manner.

## Supporting information

**S1 Appendix. CONSORT 2010 checklist for randomized trial.**
(DOCX)

**S1 Data. Dataset of the study.**
(XLSX)

**S1 Table. Comparison of baseline characteristics between participants who had completed the study and participants who had defaulted follow up (N = 1328).**
(DOCX)

**S2 Table. Baseline characteristics of study participants by clinics (N = 1200).**
(DOCX)

**S3 Table. Effects of intervention by routine e-EQ5D5L measurement on changes in patient-reported outcomes from baseline to 3, 6 and 12-month follow-up by GEE with cluster robust standard errors (N = 1200).**
(DOCX)

**S4 Table. Prevalence of patient-reported treatment and service utilization throughout the study period by groups (N = 1200).**
(DOCX)

## Acknowledgments

We acknowledge the EuroQol Group for their permission to use the electronic EQ-5D-5L (Hong Kong Chinese) questionnaire for our study and the financial support to cover the open access publication fee. We thank all the patients who participated in our study. We would also like to thank all the doctors and staff in the participating primary care clinics for their help in patient recruitment and data collection. Thanks also go to our research assistants (Ms. Au Natalie, Ms. Chan Clara, Mr. Lai Anson, Ms. Pan Rainnie, Mr. Wong Simon and Ms. Zheng Alice) for their help in data collection.

## Author contributions

**Conceptualization:** Will H.G. Cheng, Amy P. P. Ng, Carlos K.H. Wong, Calvin K.L. Or, Daniel Y.T. Fong, Jason P.Y. Cheung, David V.K. Chao, Welchie W.K. Ko, Eliza L.Y. Wong, Cindy L.K. Lam.

**Data curation:** Will H.G. Cheng, Carlos K.H. Wong, Cindy L.K. Lam.

**Formal analysis:** Will H.G. Cheng.

**Funding acquisition:** Cindy L.K. Lam.

**Methodology:** Will H.G. Cheng, Amy P. P. Ng, Carlos K.H. Wong, Calvin K.L. Or, Daniel Y.T. Fong, Jason P.Y. Cheung, David V.K. Chao, Welchie W.K. Ko, Eliza L.Y. Wong, Cindy L.K. Lam.

**Project administration:** Will H.G. Cheng.

**Supervision:** Amy P. P. Ng, Carlos K.H. Wong, Calvin K.L. Or, Daniel Y.T. Fong, Jason P.Y. Cheung, David V.K. Chao, Welchie W.K. Ko, Eliza L.Y. Wong, Cindy L.K. Lam.

**Visualization:** Kiki S.N. Liu.

**Writing – original draft:** Will H.G. Cheng, Amy P. P. Ng.

**Writing – review & editing:** Will H.G. Cheng, Kiki S.N. Liu, Amy P. P. Ng, Carlos K.H. Wong, Calvin K.L. Or, Daniel Y.T. Fong, Jason P.Y. Cheung, David V.K. Chao, Welchie W.K. Ko, Eliza L.Y. Wong, Cindy L.K. Lam.

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
