## [Decision Letter · Decision Letter 0]

6 Nov 2025

Response to Reviewers'. This file does not need to include responses to any formatting updates and technical items listed in the 'Journal Requirements' section below.'. This file does not need to include responses to any formatting updates and technical items listed in the 'Journal Requirements' section below.* A marked-up copy of your manuscript that highlights changes made to the original version. You should upload this as a separate file labeled 'Revised Manuscript with Track Changes'.'.* An unmarked version of your revised paper without tracked changes. You should upload this as a separate file labeled 'Manuscript'.'. If you would like to make changes to your financial disclosure, competing interests statement, or data availability statement, please make these updates within the submission form at the time of resubmission. Guidelines for resubmitting your figure files are available below the reviewer comments at the end of this letter. We look forward to receiving your revised manuscript. Kind regards, Jan Louis Adajo NonSupport Staff - EditorialPLOS Digital Health Dukyong YoonSection EditorPLOS Digital Health Leo Anthony CeliEditor-in-ChiefPLOS Digital Healthorcid.org/0000-0001-6712-6626 **Journal Requirements:**

i. State the initials, alongside each funding source, of each author to receive each grant. For example: "This work was supported by the National Institutes of Health (####### to AM; ###### to CJ) and the National Science Foundation (###### to AM)."

2. Please ensure that your Ethics Statement is available in its entirety at the beginning of your Methods section, under a subheading 'Ethics Statement'. It must include:

1) The name(s) of the Institutional Review Board(s) or Ethics Committee(s)

2) The approval number(s), or a statement that approval was granted by the named board(s)

3) (for human participants/donors) - A statement that formal consent was obtained (must state whether verbal/written) OR the reason consent was not obtained (e.g. anonymity).

3. Please provide separate figure files in .tif or .eps format.

4. In the online submission form, you indicated that “Data shall be available upon reasonable request to the corresponding author.”.

3. Uploaded as supplementary information.

5. Some material included in your submission may be copyrighted. According to PLOS’s copyright policy, authors who use figures or other material (e.g., graphics, clipart, maps) from another author or copyright holder must demonstrate or obtain permission to publish this material under the Creative Commons Attribution 4.0 International (CC BY 4.0) License used by PLOS journals. Please closely review the details of PLOS’s copyright requirements here: PLOS Licenses and Copyright. If you need to request permissions from a copyright holder, you may use PLOS's Copyright Content Permission form.

Potential Copyright Issues:

a. S2 Appendix and S3 Appendix contains branding/a logo. We are not permitted to publish this under our CC-BY 4.0 license, even with permission. We ask that you please remove or replace it.

b. S2 Appendix and S3 Appendix contains screenshots. We are not permitted to publish these under our CC-BY 4.0 license; websites are usually intellectual property and are copyrighted. This includes peripheral graphics of the web browser such as the [X] buttons. We ask that you please remove or replace it.

**Additional Editor Comments (if provided):** The reviewers provided generally positive feedback. They noted, however, that the manuscript requires additional details and clarifications regarding the implementation and statistical analysis. Elaborating on the potential behavioral and communication mechanisms behind the improvement was also recommended.**Reviewers' Comments:** Reviewer's Responses to Questions

**Comments to the Author**

1. Does this manuscript meet PLOS Digital Health’s publication criteria? Is the manuscript technically sound, and do the data support the conclusions? The manuscript must describe methodologically and ethically rigorous research with conclusions that are appropriately drawn based on the data presented.? Is the manuscript technically sound, and do the data support the conclusions? The manuscript must describe methodologically and ethically rigorous research with conclusions that are appropriately drawn based on the data presented.

Reviewer #1: Yes

Reviewer #2: Partly

Reviewer #3: Yes

2. Has the statistical analysis been performed appropriately and rigorously?

Reviewer #1: Yes

Reviewer #2: I don't know

Reviewer #3: Yes

3. Have the authors made all data underlying the findings in their manuscript fully available (please refer to the Data Availability Statement at the start of the manuscript PDF file)?

The PLOS Data policy requires authors to make all data underlying the findings described in their manuscript fully available without restriction, with rare exception. The data should be provided as part of the manuscript or its supporting information, or deposited to a public repository. For example, in addition to summary statistics, the data points behind means, medians and variance measures should be available. If there are restrictions on publicly sharing data—e.g. participant privacy or use of data from a third party—those must be specified.requires authors to make all data underlying the findings described in their manuscript fully available without restriction, with rare exception. The data should be provided as part of the manuscript or its supporting information, or deposited to a public repository. For example, in addition to summary statistics, the data points behind means, medians and variance measures should be available. If there are restrictions on publicly sharing data—e.g. participant privacy or use of data from a third party—those must be specified.

Reviewer #1: Yes

Reviewer #2: Yes

Reviewer #3: Yes

4. Is the manuscript presented in an intelligible fashion and written in standard English?

Reviewer #1: Yes

Reviewer #2: Yes

Reviewer #3: Yes

Reviewer #1: I would like to thank the editorial team for giving me the opportunity to review this manuscript. Also appreciate the authors’ efforts in addressing an important area of clinical research related to improving patient-reported outcomes in primary care. The topic is both timely and relevant, considering the growing emphasis on integrating HRQOL measures into chronic disease management.

Abstract

1. The abstract clearly presents the study design and findings; however, the background section could be strengthened by briefly explaining why measuring HRQOL routinely is expected to improve patient outcomes.

2. The conclusion is informative but slightly overstated. It should reflect that HRQOL measurement did not improve HRQOL or pain, but was linked to improved enablement

Introduction

3. The introduction provides a clear and logical background, but it could be strengthened by explicitly highlighting the knowledge gap i.e., although HRQOL measures are known to enhance patient–doctor communication, their actual clinical impact on outcomes in primary care remains uncertain

4. The definition of patient enablement is appropriate; still, linking it more explicitly to HRQOL improvement and chronic disease self-management would make the argument more cohesive.

5. Minor language refinements e.g., “to facilitate routine HRQOL measurement in busy primary care clinics, our team developed…” instead of “to facilitate routine HRQOL measurement in busy primary care clinics our research team has developed” would improve readability.

Method

6. The study design is appropriate (cluster RCT, 12 months, single-blind), but it would be useful to explain why the cluster design was chosen and whether intra-cluster correlation was accounted for in the sample size calculation and analysis.

7. Please clarify what “single-blind” refers to participants, doctors, or outcome assessors

8. The target sample size (n=1374) is reported, but the sample size calculation formula and assumptions (effect size, power, ICC, etc.) are not shown here.

9. Randomisation at the clinic level is appropriate, but additional detail on allocation concealment and steps taken to minimize selection bias before recruitment would be helpful.

10. Since only six clinics were included, please discuss whether stratification or matching was performed to balance clinic characteristics between groups.

11. Please clarify whether doctors in the intervention group received training or guidance on how to interpret and use HRQOL reports during consultations.

12. The schedule for baseline and follow-up assessments is well defined, but consider clarifying whether the same interviewer conducted all follow-ups and whether inter-rater reliability was assessed.

13. The data collection method (telephone vs face-to-face) differs between baseline and follow-ups; discuss any potential mode effect on data consistency.

14. The use of multiple PROMs is appropriate, but the manuscript should explain whether any correction for multiple comparisons was applied to avoid Type I error inflation.

15. Clarify how missing data were handled

16. The authors should state whether analyses were intention-to-treat (ITT) or per-protocol.

Results

17. The loss to follow-up rate is acceptable, but it should be noted whether attrition was statistically compared between groups and if missing data could have introduced bias.

18. The text mentions no significant difference in PROMs at baseline, which supports group comparability; however, please specify whether these comparisons were performed using appropriate tests adjusted for clustering.

19. Clarify whether participants with both knee and back problems were analyzed as a combined group or separately.

20. Although the tables are referenced, the narrative could briefly report key numerical values (e.g., mean changes and confidence intervals) rather than relying mainly on p-values.

21. Effect sizes and clinical significance should be discussed, particularly for WOMAC and PEI-2 outcomes.

22. The authors mention using GEE models, but it would be useful to specify whether they accounted for clustering by clinic and adjusted for baseline covariates.

23. The section could be streamlined by avoiding repetitive mentions of significance for each follow-up.

Discussion

24. The discussion could be strengthened by more explicitly addressing the clinical relevance of the observed improvements in enablement scores for instance, whether the magnitude of change in PEI-2 is meaningful in real-world practice.

25. While the authors suggest that HRQOL reporting may improve doctor awareness of psychosocial issues, this point could be expanded with reference to behavioral or patient engagement theories e.g., patient activation, shared decision-making. This would enhance the conceptual grounding of the discussion.

26. The limitations section is concise but could be expanded. Specifically, the impact of COVID-19 restrictions on both recruitment and outcomes deserves more emphasis, as it may have introduced systematic bias or affected patient behaviors differently across clusters.

27. The comparison with prior studies could benefit from a deeper discussion on why previous interventions showed improvements in mental health outcomes and why this trial did not translate that effect into HRQOL or pain reduction.

28. Future research directions could be strengthened by suggesting mixed-methods evaluations or cost-effectiveness analyses to determine the broader value of routine HRQOL measurement.

Conclusion

29. A closing remark highlighting the potential policy or primary care implications e.g., supporting personalized management plans would enhance the practical significance of the work.

Reviewer #2: Summary

This manuscript describes a cluster randomised controlled trial investigating the impact of routine electronic HRQOL measurement (using the e-EQ-5D-5L) on patient-reported outcomes in primary care patients with chronic knee/back problems. The study found that the intervention did not improve the primary outcome of disease-specific HRQOL (WOMAC) or pain but was associated with a statistically significant improvement in patient self-care enablement (PEI-2). The overall reporting quality is good, but several key items require clarification. Authors should address these.

Major points

1. The manuscript should explicitly state whether there were any changes to the trial methods or primary/secondary outcomes after commencement, relative to the published protocol. No information is provided on whether there were any changes to methods after trial commencement (e.g., to eligibility criteria, outcomes) from the published protocol.

2. The planned sample size is mentioned, but the justification (power, alpha, effect size) is not provided in the manuscript. The parameters used for the sample size calculation (alpha, power, effect size, ICC) should be provided.

3. The manuscript states randomisation was performed at clinic level by a statistician not involved in recruitment, but sequence generation method, allocation concealment, and whether stratification or restricted randomisation were used are not specified.

4. Only six clusters (3 intervention / 3 control) were randomised. With so few clusters, standard large-sample GEE variance estimates may be unreliable and type I error rates can be inflated. The manuscript does not discuss how the small number of clusters was handled in sample size calculation, analysis, or degrees-of-freedom adjustments.

5. Many baseline variables are presented at individual level (Table 1). Because randomisation occurred at cluster level, cluster-level baseline summaries and tests (or at least ICCs and cluster means) should be reported. The manuscript notes some baseline differences in healthcare use (e.g., oral prescribed medications, self-medication) which could confound post-randomisation management decisions.

6. Settings are described, but more detail on the type of primary care clinics (e.g., patient demographics, urban/rural) would be helpful

7. A statement on the monitoring and reporting of any adverse events is required, even if none occurred.

8. Attrition is reported (10.6% IG, 8.0% CG). There is no clear description of how missing data were handled (complete-case vs. imputation), whether missingness was balanced, or whether intention-to-treat (ITT) principle at the individual level was followed.

9. For the positive finding on patient enablement (PEI-2), a discussion of the Minimal Clinically Important Difference (MCID) would help interpret the practical significance of the result.

10. Primary outcome was WOMAC (range 0–96). The observed adjusted difference at 12 months was β=2.43 (worse in IG) and p=0.018. It is not clear whether this difference is clinically meaningful; authors do not discuss minimal clinically important difference (MCID) for WOMAC or place effect sizes in clinical context.

11. The Discussion is clear but limited in scope. It would benefit from a more comprehensive analysis of the findings within a broader clinical and digital-health context. At present, it mainly restates results without sufficiently exploring mechanisms, clinical relevance, or implications for practice. I recommend the authors substantially expand this section to:

- Explain why EQ-5D-5L feedback may have improved patient enablement but not pain or function, considering possible behavioural and communication mechanisms.

- Interpret the magnitude of change in WOMAC relative to the established minimal clinically important difference (MCID) to clarify whether statistically significant differences are also clinically meaningful.

-Acknowledge and discuss key methodological constraints, including the very small number of clusters, potential baseline imbalances, possible contamination, the limited generalisability beyond Hong Kong public primary care, and the appropriateness of using WOMAC for participants with back pain.

- Compare the results with previous trials or systematic reviews on digital or electronic PROM feedback interventions in primary care

- Discuss how PROM feedback could be integrated into clinical workflows, electronic health records, etc

- The current conclusion somewhat overstates the benefits of the intervention. It should be more balanced.

Reviewer #3: Reviewer Recommendation and Comments for Manuscript Effectiveness of Routine Measurement of Health-Related Quality of Life (HRQOL) in Improving Patient-reported Outcomes in Primary Care Patients with Chronic Knee and Back Problems – A Cluster Randomised Controlled Trial

The article presents the effectiveness of routine measurement of HRQOL in improving patient-reported outcomes in primary care patients with chronic knee and back problemts. The authors draw on existing literature and research from the field of HRQOL research and present the results clearly and concisely.

The abstract and Background section is well written. The Background section provides relevant and necessary information to the study and an insight in relevant previous research. The identified research gap and what this study adds to the existing body of knowledge is described and discussed. The context is described and is helpful in understanding the study purpose.

The authors provide new insights into how routine measurement of HRQOL may improve self-care and widen the range of treatments of chronic knee and back problems. The analysis and comparisions beween the IG ang CG are sufficiently described, detailed and clearly presented. The longitudinal design provide additional insights into development of the conditions and HRQOL.

**Do you want your identity to be public for this peer review?** For information about this choice, including consent withdrawal, please see our Privacy Policy..

Reviewer #1: No

Reviewer #2: No

Reviewer #3: **Yes:** Ellen Solstad OlavesenEllen Solstad OlavesenEllen Solstad OlavesenEllen Solstad Olavesen

**Figure resubmission:** While revising your submission, we strongly recommend that you use PLOS’s NAAS tool (https://ngplosjournals.pagemajik.ai/artanalysis) to test your figure files. NAAS can convert your figure files to the TIFF file type and meet basic requirements (such as print size, resolution), or provide you with a report on issues that do not meet our requirements and that NAAS cannot fix.

**Reproducibility:** To enhance the reproducibility of your results, we recommend that authors of applicable studies deposit laboratory protocols in protocols.io, where a protocol can be assigned its own identifier (DOI) such that it can be cited independently in the future. Additionally, PLOS ONE offers an option to publish peer-reviewed clinical study protocols. Read more information on sharing protocols at https://plos.org/protocols?utm_medium=editorial-email&utm_source=authorletters&utm_campaign=protocols To enhance the reproducibility of your results, we recommend that authors of applicable studies deposit laboratory protocols in protocols.io, where a protocol can be assigned its own identifier (DOI) such that it can be cited independently in the future. Additionally, PLOS ONE offers an option to publish peer-reviewed clinical study protocols. Read more information on sharing protocols at https://plos.org/protocols?utm_medium=editorial-email&utm_source=authorletters&utm_campaign=protocols

---

## [Decision Letter · Decision Letter 1]

16 Mar 2026

Effectiveness of Routine Measurement of Health-Related Quality of Life (HRQOL) in Improving Patient-reported Outcomes in Primary Care Patients with Chronic Knee and Back Problems – A Cluster Randomised Controlled Trial

PDIG-D-25-00766R1

Dear Dr Liu,

We are pleased to inform you that your manuscript 'Effectiveness of Routine Measurement of Health-Related Quality of Life (HRQOL) in Improving Patient-reported Outcomes in Primary Care Patients with Chronic Knee and Back Problems – A Cluster Randomised Controlled Trial' has been provisionally accepted for publication in PLOS Digital Health.

Best regards,

Patrick Wai-Hang Kwong, PhD

Academic Editor

PLOS Digital Health

**Additional Editor Comments (if provided):**

The authors have addressed all the concern from reviewers.

**Reviewer Comments (if any, and for reference):**

Reviewer's Responses to Questions

**Comments to the Author**

Reviewer #1: All comments have been addressed

Reviewer #3: All comments have been addressed

publication criteria? Is the manuscript technically sound, and do the data support the conclusions? The manuscript must describe methodologically and ethically rigorous research with conclusions that are appropriately drawn based on the data presented.? Is the manuscript technically sound, and do the data support the conclusions? The manuscript must describe methodologically and ethically rigorous research with conclusions that are appropriately drawn based on the data presented.

Reviewer #1: Yes

Reviewer #3: Yes

3. Has the statistical analysis been performed appropriately and rigorously?

Reviewer #1: Yes

Reviewer #3: Yes

4. Have the authors made all data underlying the findings in their manuscript fully available (please refer to the Data Availability Statement at the start of the manuscript PDF file)?

The PLOS Data policy requires authors to make all data underlying the findings described in their manuscript fully available without restriction, with rare exception. The data should be provided as part of the manuscript or its supporting information, or deposited to a public repository. For example, in addition to summary statistics, the data points behind means, medians and variance measures should be available. If there are restrictions on publicly sharing data—e.g. participant privacy or use of data from a third party—those must be specified.requires authors to make all data underlying the findings described in their manuscript fully available without restriction, with rare exception. The data should be provided as part of the manuscript or its supporting information, or deposited to a public repository. For example, in addition to summary statistics, the data points behind means, medians and variance measures should be available. If there are restrictions on publicly sharing data—e.g. participant privacy or use of data from a third party—those must be specified.

Reviewer #1: Yes

Reviewer #3: Yes

5. Is the manuscript presented in an intelligible fashion and written in standard English?

Reviewer #1: Yes

Reviewer #3: Yes

Reviewer #1: Thank you for the revision. This version is more appropriate

Reviewer #3: All comments have been addressed, and the manuscript is polished.

**Do you want your identity to be public for this peer review?** For information about this choice, including consent withdrawal, please see our Privacy Policy..

Reviewer #1: **Yes:** Seyed kian haji seyed javadiSeyed kian haji seyed javadiSeyed kian haji seyed javadiSeyed kian haji seyed javadi

Reviewer #3: **Yes:** Ellen Solstad OlavesenEllen Solstad OlavesenEllen Solstad OlavesenEllen Solstad Olavesen
